# Changes in Left Ventricular Ejection Fraction and Oxidative Stress after Phosphodiesterase Type-5 Inhibitor Treatment in an Experimental Model of Retrograde Rat Perfusion

**DOI:** 10.3390/medicina59030458

**Published:** 2023-02-24

**Authors:** Milos Krivokapic, Israpil Alisultanovich Omarov, Vladimir Zivkovic, Tamara Nikolic Turnic, Vladimir Jakovljevic

**Affiliations:** 1Faculty of Medicine, University of Montenegro, Kruševac bb, 81000 Podgorica, Montenegro; 2Medical and Health Center of the Ministry of Foreign Affairs of Russia, 119002 Moscow, Russia; 3Department of Physiology, Faculty of Medical Sciences, University of Kragujevac, Svetozara Markovica 69, 34000 Kragujevac, Serbia; 4Department of Clinical Pharmacology, I.M. Sechenov First Moscow State Medical University (Sechenov University), 119146 Moscow, Russia; 5Center of Excellence for Redox Balance Research in Cardiovascular and Metabolic Disorders, 34000 Kragujevac, Serbia; 6N.A. Semashko Public Health and Healthcare Department, F.F. Erismann Institute of Public Health, I.M. Sechenov First Moscow State Medical University (Sechenov University), 119146 Moscow, Russia; 7Department of Pharmacy, Faculty of Medical Sciences, University of Kragujevac, Svetozara Markovica 69, 34000 Kragujevac, Serbia; 8Department of Human Pathology, I.M. Sechenov First Moscow State Medical University (Sechenov University), 119146 Moscow, Russia

**Keywords:** tadalafil, vardenafil, oxidative stress, ejection fractioning, isolated rat heart, Langendorff technique

## Abstract

*Background and objectives*: Taking into consideration the confirmed role of oxidative stress in ischemia/reperfusion injury and the insufficiency in knowledge regarding the phosphodiesterase 5 (PDE5)-mediated effects on the cardiovascular system, the aim of our study was to investigate the influence of two PDE5 inhibitors, tadalafil and vardenafil, with or without the addition of N(G)-nitro-L-arginine methyl ester (L-NAME), on oxidative stress markers, coronary flow and left ventricular function, both ex vivo and in vivo. *Methods*: This study included 74 male Wistar albino rats divided into two groups. In the first, 24 male Wistar rats were orally treated with tadalafil or vardenafil for four weeks in order to perform in vivo experiments. In the second, the hearts of 50 male Wistar albino were excised and perfused according to the Langendorff technique in order to perform ex vivo experiments. The hearts were perfused with tadalafil (10, 20, 50 and 200 nM), vardenafil (10, 20, 50 and 200 nM) and a combination of tadalafil/vardenafil and L-NAME (30 μM). The CF and oxidative stress markers, including nitrite bioaviability (NO_2_^−^), superoxide anion radical (O_2_^−^), and the index of lipid peroxidation, were measured in coronary effluent. *Results*: The L-arginin/NO system acts as the mediator in the tadalafil-induced effects on the cardiovascular system, while it seems that the vardenafil-induced increase in CF was not primarily induced by the NO system. Although tadalafil induced an increase in O_2_^−^ in the two lowest doses, the general effects of both of the applied PDE5 inhibitors on oxidative stress were not significant. The ejection function was above 50% in both groups. *Conclusions*: Our results showed that both tadalafil and vardenafil improved the coronary perfusion of the myocardium and LV function by increasing the EF.

## 1. Introduction

Phosphodiesterase inhibitors (PDIs) selectively antagonize phosphodiesterase (PDE), which catalyzes cyclic adenosine monophosphate (cAMP) and cyclic guanosine monophosphate (cGMP), and is an important determinant in the regulation of the intracellular concentrations and biological actions of these secondary messengers. PDE5, which is cGMP-specific, is found in high abundance in a variety of human cells [1]. These phosphodiesterase inhibitors are drugs that can accumulate nitric oxide, which is formed from cGMP, and as a result, vasodilatation begins in the corpus cavernosum and vasculature among lungs. These physiological properties of PDE-5 inhibitors are a good choice in treating erectile dysfunction and pulmonary arterial hypertension [2]. As the myocardial tissue contains PDE-5, it is possible that this could also be cardioprotective [3,4]. The cardioprotective effect of PDIs has been linked with apoptosis and necrosis by using the many signaling mechanisms, such as the higher expression of nitric synthase (endothelial and inducible), the activation of the protein C and G kinases, etc. Generally, it is suspected that PDE5 is not present in normal myocytes, and the selective inhibition of type 5 PDE likely does not have an inotropic effect in physiological conditions [5]. Sildenafil (Viagra^®^), vardenafil (Levitra^®^), and tadalafil (Cialis^®^) are three major PDE5 inhibitors. Sildenafil was the first PDE5 inhibitor shown to exert a protective effect against ischemia-reperfusion injury in the hearts of rabbits, mice, rats, and dogs [6,7,8]. It is also known that sildenafil acts as a reductor of pulmonary and systemic resistance. In addition, sildenafil improves pulmonary gas diffusion and increases cardiac output. The hemodynamic effects of sildenafil are based on changing the left ventricular pressure and lead to an improvement in left ventricular function [8]. Similarly, tadalafil acts on the Beta-adrenergic signaling cascade in the heart, which is the most highly responsible mechanism of these drugs. In addition, the myocardial response to catecholamine stimulation is some kind of outcome of Beta stimulation. In this case, phosphodiesterase inhibition is highly responsible for changing the function of the myocardium [6,7,8,9].

It is well known that oxidative stress impairs the vasodilation of the coronary, pulmonary, and peripheral vasculature [9]. The state of oxidative stress as an imbalance of pro-oxidants and antioxidants in the body is the key mechanism underlying myocardial infarction. Free radicals can be divided into two categories: reactive oxygen species (ROS) and reactive nitrogen species (RNS). Additionally, enzymatic antioxidant defense, including superoxide dismutase, glutathione peroxidase and catalase activity, is very important for maintaining health [10]. Thus, when the content of ROS and RNS in an organism exceeds the scavenging ability of its own antioxidant capacity, this can destroy the redox balance and trigger various areas of damage in different organs [10].

However, there is a lack of information regarding the effect of PDE5 inhibitors on endothelial oxidative status. Previous studies have suggested that sildenafil could prevent oxidative stress by reducing the free radicals and improving the antioxidant enzyme systems [10]. On the other hand, vardenafil could reduce the formation of 3-nitrotyrosine and increase the bioavailability of nitric oxide in rat models and human studies [11]. The balance between the production and removal of cGMP is an important regulator of coronary blood flow [12]. Through NO/cGMP-dependent pathways, vardenafil relaxes the resistance in coronary vessels [13].

While these effects have been widely reproduced, a smaller number of studies have attested a similar cardioprotective effect of vardenafil and tadalafil [14]. Taking into consideration the confirmed role of oxidative stress in cardiac homeostasis and the insufficient knowledge about the PDE5-mediated effects on the cardiovascular systems, the aim of our study was to investigate the influence of two PDE5 inhibitors, tadalafil and vardenafil, with or without the addition of N(G)-nitro-L-arginine methyl ester (L-NAME), on oxidative stress markers, coronary flow, and left ventricular function, ex vivo and in vivo.

## 2. Methods

### 2.1. Ethical Concerns

All research procedures were carried out in accordance with the EU Directive 2010/63/EU for animal experiments and the principles of Good Laboratory Practice (GLP), approved by the ethical committee of the Faculty of Medical Sciences, University of Kragujevac, Serbia and Ministry of Health, Republic of Serbia.

### 2.2. Animals

This study included 74 male Wistar albino rats divided into two groups. In the first, 24 male Wistar rats were orally treated with tadalafil or vardenafil for 4 weeks in order to perform in vivo experiments. In the second, the hearts of 50 male Wistar albino were excised and perfused according to the Langendorff technique in order to perform ex vivo experiments. The animals were housed under standard controlled environmental conditions, with a temperature of 23 ± 1 °C and a 12/12 h light/dark cycle. Food and water were provided ad libitum. All animals were obtained from the Military Medical Academy (Belgrade, Serbia).

### 2.3. Two-Dimensional Echocardiography of Rat Heart In Vivo

In the two groups of male Wistar albino rats (n = 24; 8 weeks old, 200 ± 30 g) who were treated with tadalafil and vardenafil (20 mg/bw/day per os) for four weeks, the echocardiography method was used for evaluating the LV function in vivo. The transthoracic 2D echocardiography (EHO) evaluation was carried out using a Hewlett-Packard Sonas 55000 (Palo Alto, CA, USA) sector scanner furnished with an MHz phased-array transducer. The rats were anesthetized by an intraperitoneal injection mixture of 10 mg/kg ketamine (0.025 mL of 100 mg/mL, Pfizer Pharmaceuticals, New York, NY, USA) and 2 mg/kg xylazine (0.025 mL of 20 mg/mL, Intechemie, Waalre, Holland) [15]. The following structural variables were measured:Interventricular septal wall thickness at end-diastoles (IVSds) and end-systoles (IVSs);Left ventricle (LV) internal dimension at end-diastoles (LVIDds) and end-systoles (LVIDs);LV posterior wall thickness at end-diastoles (LVPWds) and end-systoles (LVPWs);Fractional shortening (FS) percentage.

On M-mode tracking, the average values were derived from a minimum of five cardiac sets. The ejection fraction (EF) values were calculated according to the Teicholz formula:EF = 100 × (LVEDV − LVESV)/LVEDV
LVESV = 7 × LVESD/(2.4 + LVESD); LVEDV = 7 × LVEDD/(2.4 + LVEDD)

### 2.4. Retrograde Perfusion of Rat Heart

From fifty male Wistar albino rats (aged 8 weeks, weight circa 200 g), the hearts were isolated according to the Langendorff technique [16]. Langendorff apparatus is a system for retrograde perfusion ex vivo, purchased from Experimetria Ltd., Budapest, Hungary, which mimics the physiological conditions of heart pumping. After using general anesthesia (ketamine/xylazine), the rats were sacrificed, and using surgical intervention, the rat hearts were isolated. Immediately after that, the aortas were cannulated and the hearts were perfused with Krebs–Henseleit solution, which contains the following substances in mmol/1: NaCl 118, KCI 4.7, CaCI_2_ • _2_H_2_O 2.5, MgSO_4_ • 7H2O 1.7, NaHCO_3_ 25, KH_2_PO_4_ 1.2, glucose 11, and pyruvate 2, equilibrated with 95% O_2_ plus 5% CO_2_ and warmed to 37 °C (pH 7.4).

After a period of stabilization for 30 min, perfusion was performed at gradually increasing pressures, from 70 to 120 cm H_2_O. Coronary venous effluent was continuously collected (mL/min) and measured twice for each pressure of the water column [16]. The hearts were perfused with the PDE5 inhibitors, tadalafil (PubChem CID 110635) (10, 20, 50, 200 nM) and vardenafil (PubChem CID 110634) (10, 20, 50, 200 nM), separately or with an inhibitor of nitric oxide synthesis, N(G)-nitro-L-arginine methyl ester (PubChem CID 39836) (L-NAME, 30 μM, minimum 5 min), and compared with the controls, respectively. The testing of the substances was started after the control perfusion and after stabilization for a minimum of 5 min at each point of coronary perfusion pressure.

### 2.5. Measuring of Biomarkers of Oxidative Stress

After collecting the effluent, spectrophotometric determination was used to measure the various markers of oxidative stress, such as nitrites, superoxide anion radicals, and the index of lipid peroxidation. Krebs-Henseleit was always used as a blank probe. All biochemical parameters were determined spectrophotometrically using a Shimadzu UV 1800 spectrophotometer (Kyoto, Japan).

#### 2.5.1. Nitrite (NO_2_^−^) Determination

The determination of nitric oxide was performed by the indirect method of measuring the levels of nitrite in an effluent. Griess reagents, perfusate, and sulpho-salicylic acid were used for the procedure of determination and measured at 543 nm [15].

#### 2.5.2. Superoxide Anion Radical Determination (O_2_^−^)

This biomarker of oxidative stress was determined using Nitro blue Tetrazolium and its reaction with TRIS buffer measured at 530 nm [16].

#### 2.5.3. Index of Lipid Peroxidation (Thiobarbituric Acid Reactive Substances, TBARS)

The concentration of lipid peroxidation in the coronary venous effluent was measured through the estimation of thiobarbituric acid reactive substances (TBARS) using 1% TBA (Thiobarbituric Acid) in 0.05 NaOH incubated with coronary effluent at 100 °C for 15 min and read at 530 nm. Krebs-Henseleit solution was used as a blank probe [17].

### 2.6. Statistical Analysis

Values are expressed as the mean ± standard error of mean. Two well-known tests of normality, the Kolmogorov-Smirnov test and the Shapiro-Wilk test, were the methods used to test the normality of the data. For testing the differences between groups in each parameter, two-way ANOVA or Student’s *t* test was used to confirm the differences. Normality tests and analytical tests were conducted using the statistical software SPSS version 26. The statistical significance threshold was set at 0.05.

## 3. Results

### 3.1. LV Function using Echocardiography In Vivo

This part of the study evaluates the effects of the different PDE-5 inhibitors (tadalafil and vardenafil) on LV function by ultrasound evaluation (Table 1). In the comparison of these two groups, we observed similar effects on the LV function and all tested parameters. The EF was not significantly different (Table 1).

### 3.2. Coronary Flow (CF)

The coronary flow (CF) increased proportionally to the coronary perfusion pressure in the entire range of perfusion pressures studied in both the control and study groups. During the control conditions, the CF varied in a range between 3.14 ± 0.60 mL/min/g wt at 40 cm H_2_O and 10.40 ± 0.92 mL/min/g wt at 120 cm H_2_O (average values for all eight experimental protocols—Figure 1 and Figure 2). Tadalafil induced a significant increase in the CF in all applied doses (Figure 1A–D), while vardenafil induced similar changes when applied in concentrations of 20 and 200 nM (Figure 2B,D). The simultaneous application of L-NAME significantly reduced the influence of both inhibitors on the CF (Figure 1E–H and Figure 2E–H).

### 3.3. Nitrite Outflow (NO_2_^−^)

Under the control conditions, the nitrite outflow varied between 4.22 ± 1.62 mL/min/g wt at 40 cm H_2_O and 14.36 ± 2.92 mL/min/g wt at 120 cm H_2_O, which was parallel with the CPP–CF curve (average values for all eight experimental protocols—Figure 3 and Figure 4). Tadalafil-induced changes in the CF were accompanied by a significant increase in the nitrite outflow in all applied doses, except in the dose of 10 nM (Figure 3A–D). On the other hand, vardenafil-induced changes in the CF were accompanied by non-significant changes in the nitrite outflow in the first three doses (Figure 4A–C), while 200 nM of vardenafil significantly reduced the nitrite outflow in the isolated rat hearts (Figure 4D). The simultaneous application of L-NAME abolished the effects of tadalafil on the nitrite outflow (Figure 3E–H). In contrast, L-NAME did not significantly change the primary effect of vardenafil on the nitrite outflow (Figure 4E–H).

### 3.4. Superoxide Anion Production (O_2_^−^)

Under the control conditions, the superoxide anion outflow varied between 16.88 ± 4.53 mL/min/g wt at 40 cm H2O and 40.71 ± 11.32 mL/min/g wt at 120 cm H_2_O and was parallel with the CPP–CF curve (average values for all eight experimental protocols—Figure 5 and Figure 6). Tadalafil- and vardenafil-induced changes in CF were accompanied by a significant increase in the superoxide anion production at all of the applied doses (Figure 5B–D and Figure 6A–D), except when these drugs were applied at a dosage of 50 nM (Figure 5C and Figure 6C). The simultaneous application of L-NAME did not induce any significant changes in the tadalafil-treated hearts (Figure 6E–H). In contrast, L-NAME induced a significant reduction in the superoxide anion production at doses of 20 and 200 nM (Figure 6E–H).

### 3.5. Index of Lipid Peroxidation (TBARS Production)

Under the control conditions, the TBARS outflow varied between 3.25 ± 1.53 μL/min/g wt at 40 cm H_2_O and 5.23 ± 2.16 mL/min/g wt at 120 cm H_2_O and was parallel with the CPP-CF curve (average values for all eight experimental protocols—Figure 7 and Figure 8). Tadalafil-induced changes in the CF were accompanied by a significant increase in the TBARS production at doses of 20 and 200 nM (Figure 7B,D), while the other two doses did not induce significant changes in the TBARS production (Figure 7A,C). Furthermore, vardenafil did not induce significant changes in the TBARS production at any of the applied doses (Figure 8A–D). The simultaneous application of L-NAME did not induce any significant changes in the TBARS production in either the tadalafil- or vardenafil-treated hearts (Figure 7E–H and Figure 8E–H) compared to the controls.

## 4. Discussion

The aim of our study was to assess the effects of two selective and highly affinitive PDE5 (tadalafil and vardenafil) inhibitors on the oxidative stress markers in isolated rat hearts as there is currently not enough information on these effects. Additionally, using an in vivo model, we treated rats with therapeutic doses of these drugs in order to compare their effects on LV function. In order to exclude the contribution of the L-arginin/NO system to these effects, we also performed additional experiments that consisted of the simultaneous application of a PDE5 inhibitor and L-NAME. The study design and applied doses in our work are very similar to the ones published by du Toit and coworkers, who investigated the effects of the same doses of sildenafil on the infarct size in experimental acute myocardial infarction [17].

Tadalafil and vardenafil are two of the most widely used PDE5 inhibitors in the management and treatment of chronic obstructive pulmonary disease, erectile dysfunction, pulmonary hypertension, and heart failure [17]. The main molecular mechanism is based on preventing cAMP and cGMP degradation, which induces many effects, such as muscle relaxation, vasodilatation, and bronchodilatation [18]. Both treatments could be used either as a monotherapy or in combination with other agents in relation to the condition. In the heart, the precise mechanism is based on increasing the ionized calcium, vasodilating the peripheral vessels, and preventing the platelet aggregation [19]. At the endothelial level, PDE5 inhibitors induce the realization of the nitric oxide, which relaxes the vessels in the corpus cavernosum. Previous research has suggested that these inhibitors could reduce inflammation, cell proliferation, and blood viscosity [19]. As there is a lot of vasoconstriction in heart failure, NO release is one of the major factors involved in improving heart function. Inhibitors of PDE5 exert their very important effects in the heart and myocardium, and their role can be very significant in left or right ventricular hypertrophy, or congestive heart failure [20].

The application of tadalafil induced a significant increase in the coronary flow at all applied doses. On the other hand, vardenafil induced similar effects only in two doses—the middle one (20 nM) and the highest one (200 nM)—which suggests that there is not a dose-dependent effect of this PDE5 inhibitor. When those PDE5 inhibitors were applied with the addition of L-NAME, their previously expressed effects on the coronary flow were abolished. This finding suggests an important role of the L-arginin/NO system in the PDE5 inhibitor-induced changes in the coronary flow in the isolated rat hearts. This effect is in accordance with the previously reported interaction between sildenafil and L-NAME in severe hypertension and myocardial ischemia-reperfusion injury [20].

Tadalafil-induced changes in the coronary flow were accompanied by a significant increase in the nitrite outflow in all applied doses, except in the smallest one. On the other hand, vardenafil did not induce significant changes in the nitrites unless administrated at the highest dose. The simultaneous application of L-NAME reversed the effects of tadalafil, while there were no significant changes in the case of vardenafil. This supports the hypothesis regarding the contribution of the L-arginin/NO system on the effects of tadalafil and suggests that there may be another endothelial pathway for vardenafil-induced vascular effects [21,22,23,24,25,26,27,28,29]. It seems that tadalafil expresses its effects similarly to sildenafil, through the augmentation of NO synthase and cGMP levels, which points to the possibility that the cardiovascular dysfunctions that include decreased NOS activity may be alleviated by tadalafil treatment [30,31].

The changes seen in the extent of superoxide anion radical production after administration of 10 and 20 nM of tadalafil and the absence of these changes when tadalafil was applied with L-NAME point to the role of nitric oxide synthase in the observed increase in free radical production. The results regarding the effects of vardenafil and vardenafil + L-NAME on superoxide anion radical production again suggest that there is not a possibility to establish the rule according to which vardenafil expresses its effects. The fact that the application of tadalafil in combination with L-NAME abolished the effects of tadalafil, seen in both the superoxide anion radical and nitric oxide, confirms the hypothesis regarding the role of nitric oxide synthase in the tadalafil-induced increase in oxidative stress. The effects of tadalafil on the oxidative stress markers found in our study are not in agreement with the results obtained by some other authors. In one study, tadalafil decreased the levels of hydrogen peroxide, while in two others, sildenafil expressed the same effects on both the superoxide anion radical and hydrogen peroxide [21,22,23]. However, it should be taken into account that these studies were performed on men with erectile dysfunction and models with heart failure [24,25,26,27]. In both these cases, the basic antioxidant system was seriously damaged, which was not the case in our study.

The index of lipid peroxidation did not follow the dynamics observed in the above-discussed parameters following tadalafil administration. It seems that only 20 nM of tadalafil induced a similar increase in the oxidative stress parameters, which was abolished by NO releasing. The other applied doses of both inhibitors did not significantly influence the oxidative stress. The increase in the index of lipid peroxidation may be the consequence of increased nitric oxide production, which is in accordance with our previous data [28,29]. Elrod et al. observed the reduction in myocardial ischemia/reperfusion injury by sildenafil in an animal model, and that the reduction is induced by nitric oxide release [32].

Taken together with the hemodynamics examination, we can conclude that PDE-5 inhibitors, such as tadalafil and sildenafil, act similarly on cardiac function. Although there were mild changes in the oxidative stress, we have concluded that from a functional perspective, there were no significant differences affecting the LV function. This is expected because PDE5 inhibitors can cause nitrate-like hemodynamic effects, lowering wedge pressure, pulmonary dilatation, etc. This makes them useful for treating a wide range of medical conditions, now with the knowledge that oxidative stress reduction is the underlying mechanism.

Finally, it is important to emphasize that the limitations of this study could have contributed to the absence of heart rate and contractility measurements, as well as cGMP concentrations (which could more clearly show interaction between the L-arginine/NO system and PDE5). However, taken together, this is one of the first experimental studies to present both ex vivo and in vivo results of heart function after sildenafil and vardenafil treatment.

## 5. Conclusions

In summary, our results clearly show that both tadalafil and vardenafil improved the coronary perfusion of the myocardium. Those effects were NO-mediated in the tadalafil-treated group. On the contrary, the vardenafil-induced increase in the coronary flow was not primarily induced by the NO system. Finally, whilst tadalafil induced an increase in the superoxide anion radical at the two lowest doses, neither of the applied PDE5 inhibitors disturbed the redox status of the heart. According to the in vivo experiment, tadalafil and vardenafil definitely affect the left ventricular function and ejection fraction similarly.

## Figures and Tables

**Figure 1 medicina-59-00458-f001:**
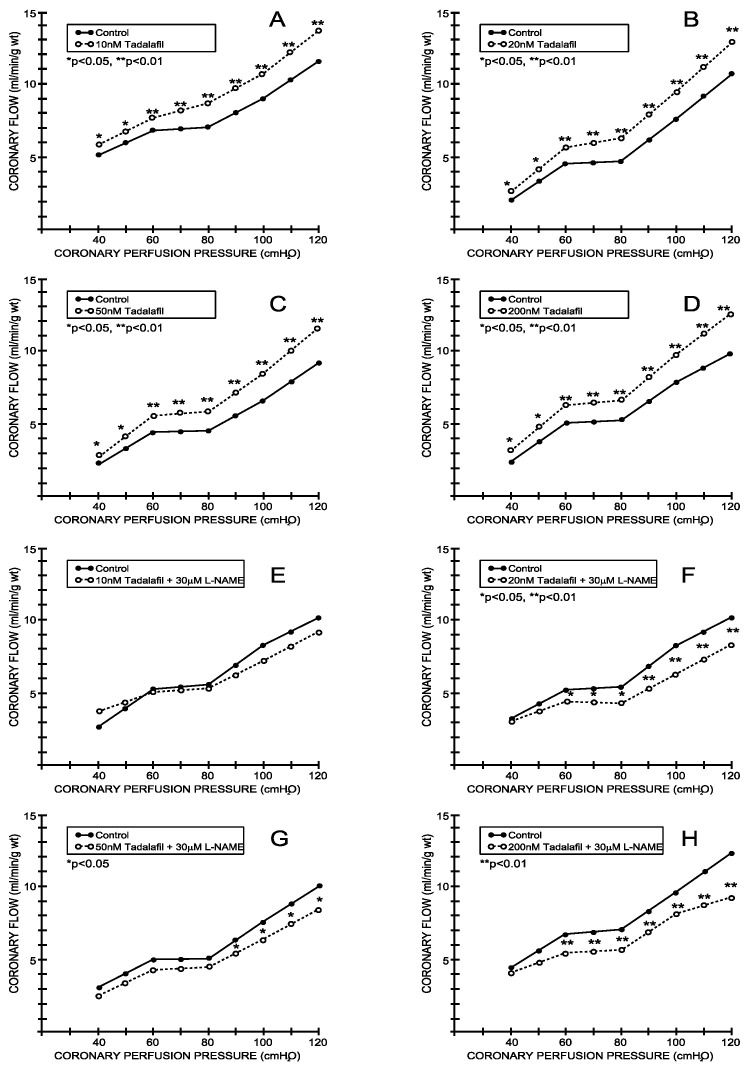
Effects of tadalafil and combination tadalafil + L-NAME on coronary flow. (**A**) Tadalafil, 10 nM, (**B**) 20 nM, **C**. 50 nM, (**D**) 200 nM, (**E**) Tadalafil + L-NAME, 10 nM + 30 μM, (**F**) Tadalafil + L-NAME, 20 nM + 30 μM, (**G**) Tadalafil + L-NAME, 50 nM + 30 μM, (**H**) Tadalafil + L-NAME, 200 nM + 30 μM.

**Figure 2 medicina-59-00458-f002:**
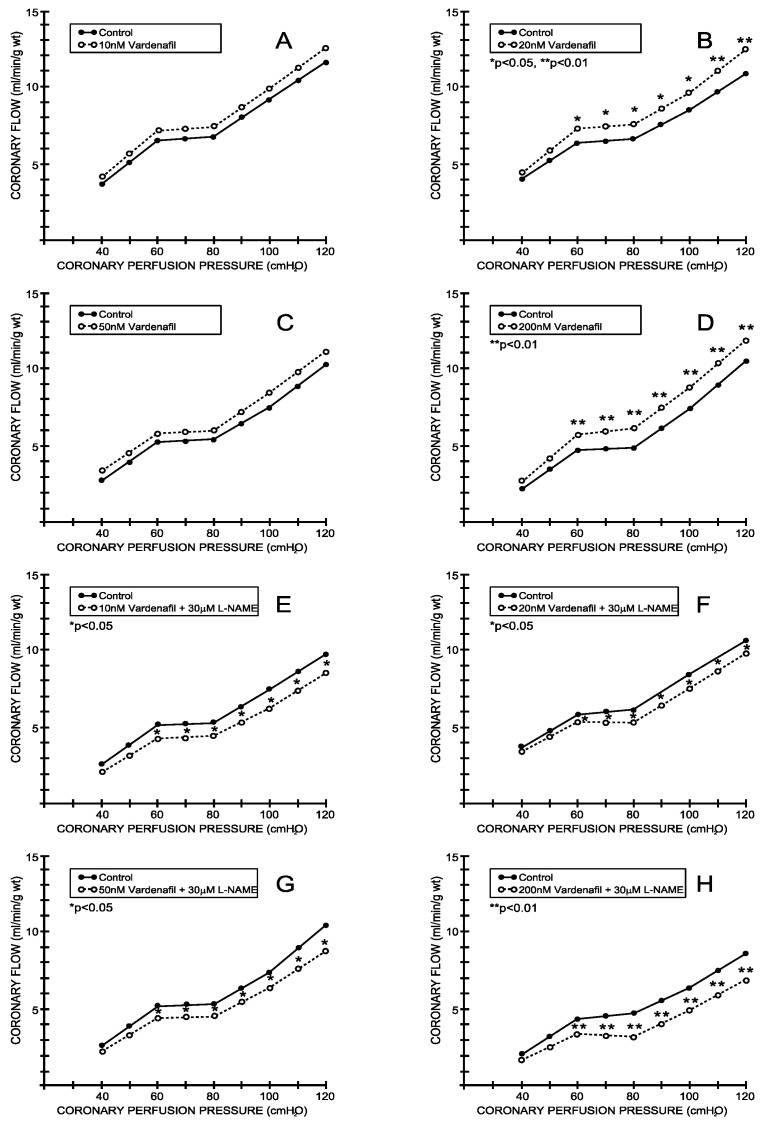
Effects of vardenafil and combination vardenafil + L-NAME on coronary flow. (**A**) Vardenafil, 10 nM, (**B**) 20 nM, (**C**) 50 nM, (**D**) 200 nM, (**E**) Vardenafil + L-NAME, 10 nM + 30 μM, (**F**) Vardenafil + L-NAME, 20 nM + 30 μM, (**G**) Vardenafil + L-NAME, 50 nM + 30 μM, (**H**) Vardenafil + L-NAME, 200 nM + 30 μM.

**Figure 3 medicina-59-00458-f003:**
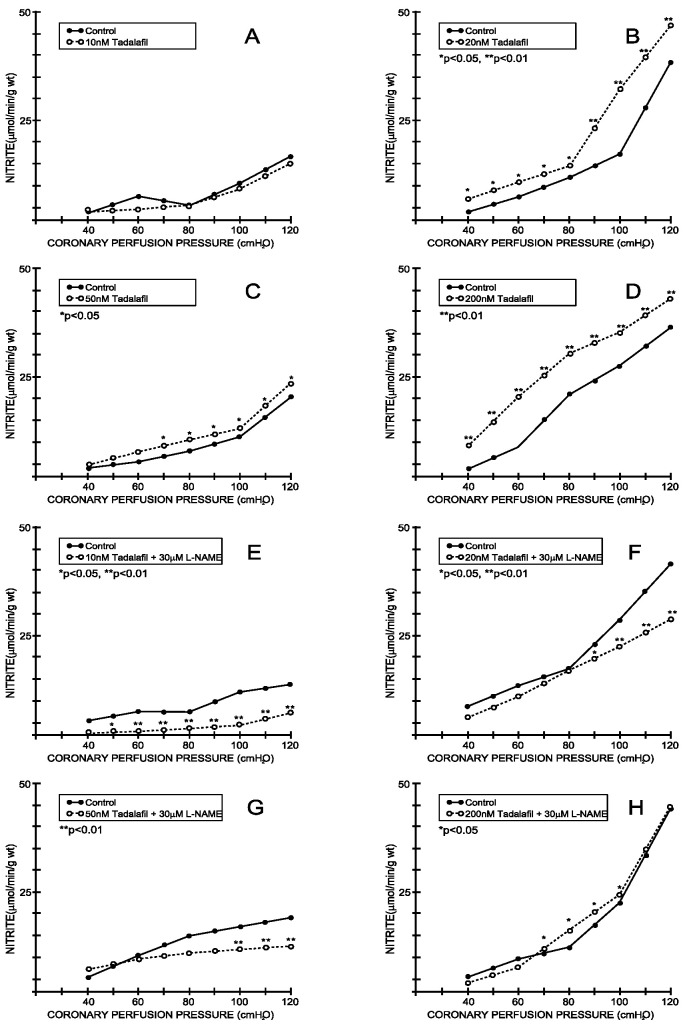
Effects of tadalafil and combination tadalafil + L-NAME on nitrite outflow. (**A**) Tadalafil, 10 nM, (**B**) 20 nM, (**C**) 50 nM, (**D**) 200nM, (**E**) Tadalafil + L-NAME, 10 nM + 30 μM, (**F**) Tadalafil+ L-NAME, 20 nM + 30 μM, (**G**) Tadalafil + L-NAME, 50 nM + 30 μM, (**H**) Tadalafil + L-NAME, 200 nM + 30 μM.

**Figure 4 medicina-59-00458-f004:**
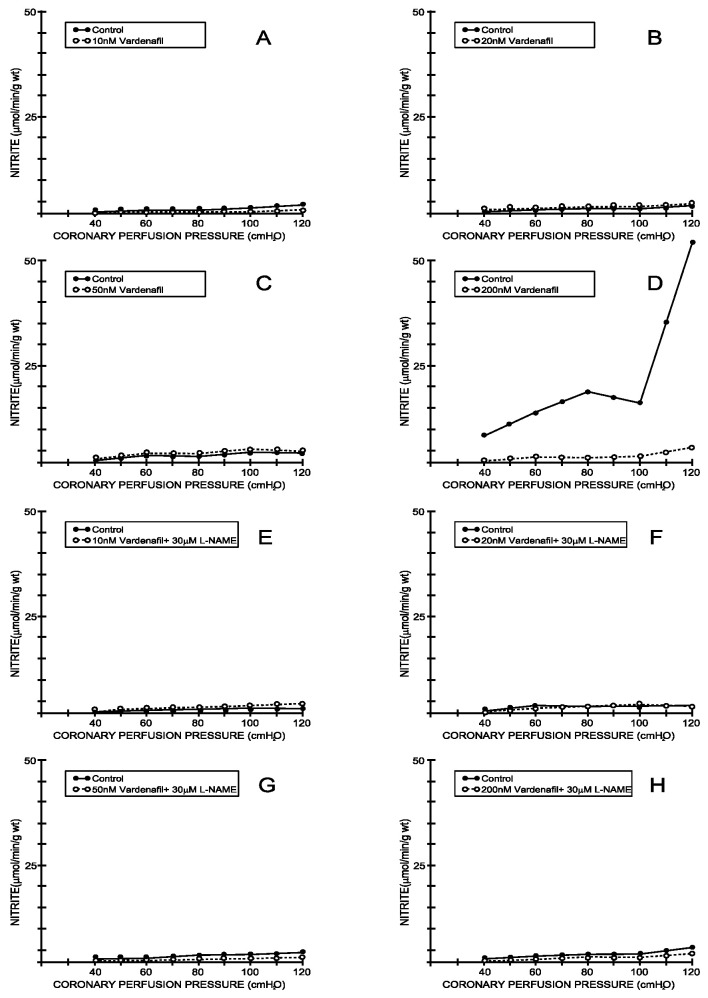
Effects of vardenafil and combination vardenafil + L-NAME on nitrite outflow. (**A**) Vardenafil, 10 nM, (**B**) 20 nM, (**C**) 50 nM, (**D**) 200 nM, (**E**) Vardenafil + L-NAME, 10 nM + 30 μM, (**F**) Vardenafil + L-NAME, 20 nM + 30 μM, (**G**) Vardenafil + L-NAME, 50 nM + 30 μM, (**H**) Vardenafil + L-NAME, 200 nM + 30 μM.

**Figure 5 medicina-59-00458-f005:**
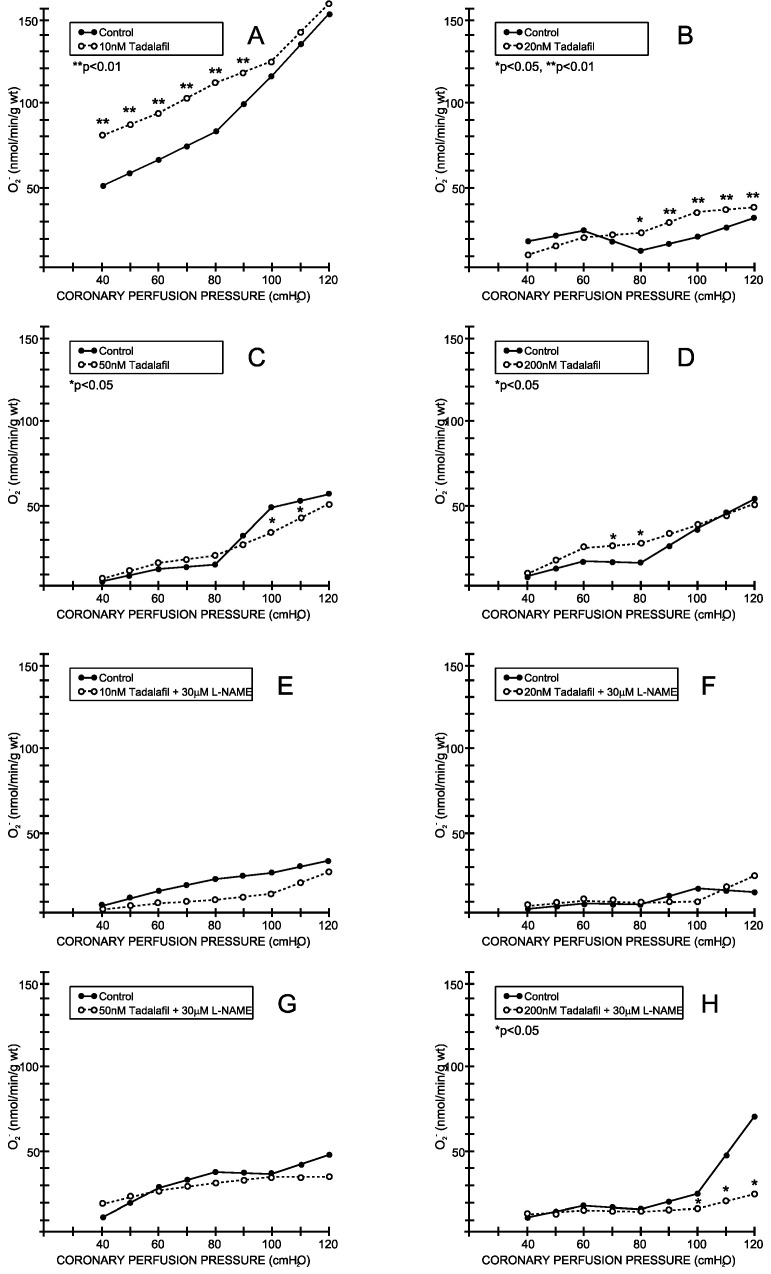
Effects of tadalafil and combination tadalafil + L-NAME on superoxide anion radicals. (**A**) Tadalafil, 10 nM, (**B**) 20 nM, (**C**) 50 nM, (**D**) 200 nM, (**E**) Tadalafil + L-NAME, 10 nM + 30 μM, (**F**) Tadalafil + L-NAME, 20 nM + 30 μM, (**G**) Tadalafil + L-NAME, 50 nM + 30 μM, (**H**) Tadalafil + L-NAME, 200 nM + 30 μM.

**Figure 6 medicina-59-00458-f006:**
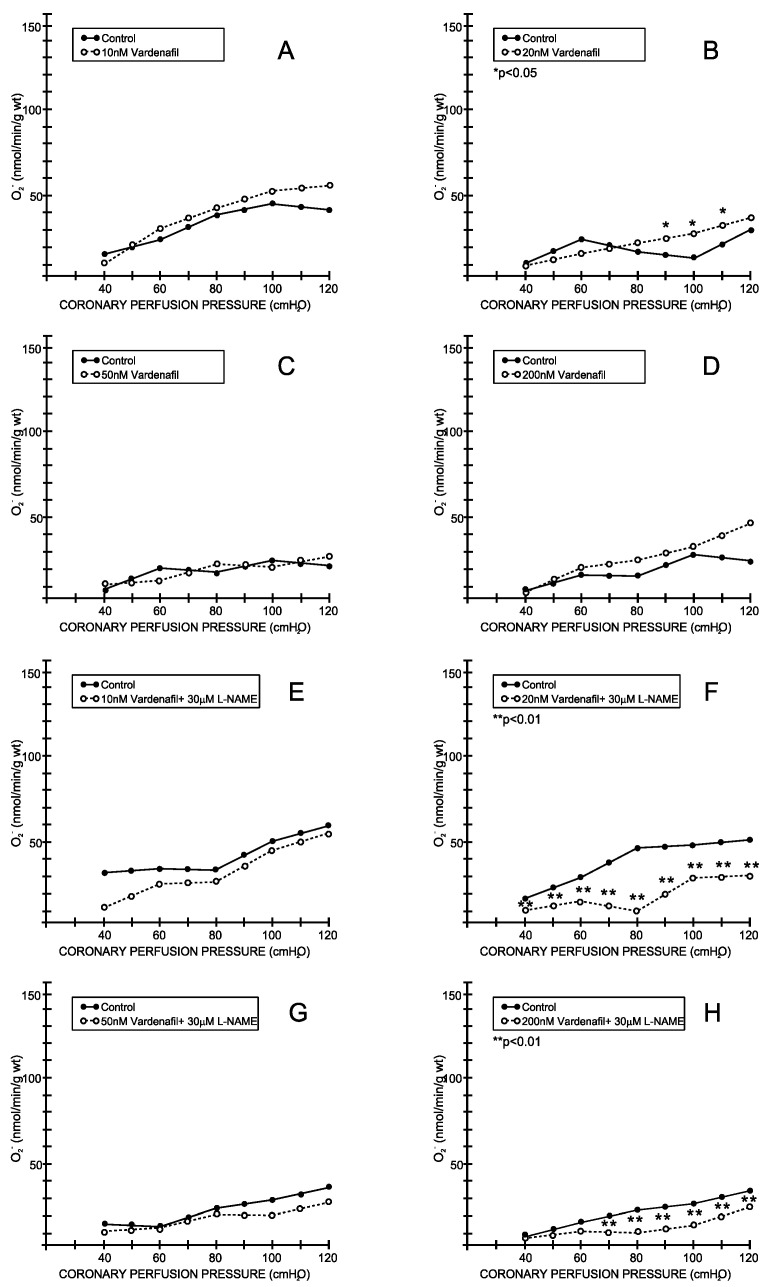
Effects of vardenafil and combination vardenafil + L-NAME superoxide anion radicals. (**A**) Vardenafil, 10 nM, (**B**) 20 nM, (**C**) 50 nM, (**D**) 200 nM, (**E**) Vardenafil + L-NAME, 10 nM + 30 μM, (**F**) Vardenafil + L-NAME, 20 nM + 30 μM, (**G**) Vardenafil + L-NAME, 50 nM + 30 μM, (**H**) Vardenafil + L-NAME, 200 nM + 30 μM.

**Figure 7 medicina-59-00458-f007:**
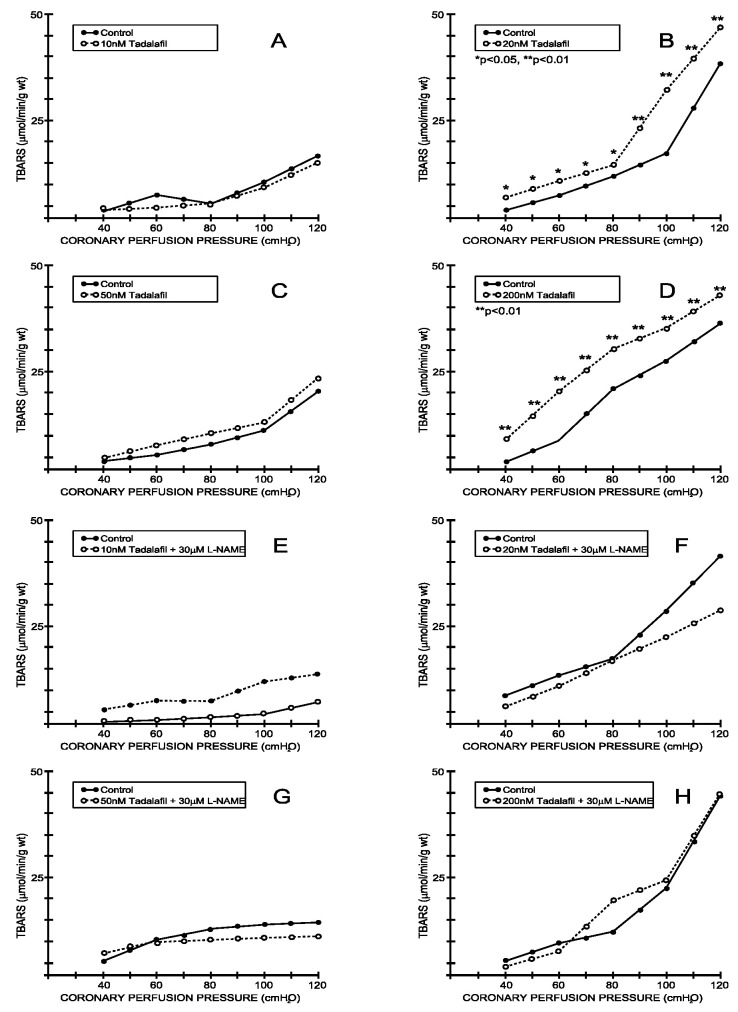
Effects of tadalafil and combination tadalafil + L-NAME on index of lipid peroxidation. (**A**) Tadalafil, 10 nM, (**B**) 20 nM, (**C**) 50 nM, (**D**) 200 nM, (**E**) Tadalafil + L-NAME, 10 nM + 30 μM, (**F**) Tadalafil + L-NAME, 20 nM + 30 μM, (**G**) Tadalafil + L-NAME, 50 nM + 30 μM, (**H**) Tadalafil + L-NAME, 200 nM + 30 μM.

**Figure 8 medicina-59-00458-f008:**
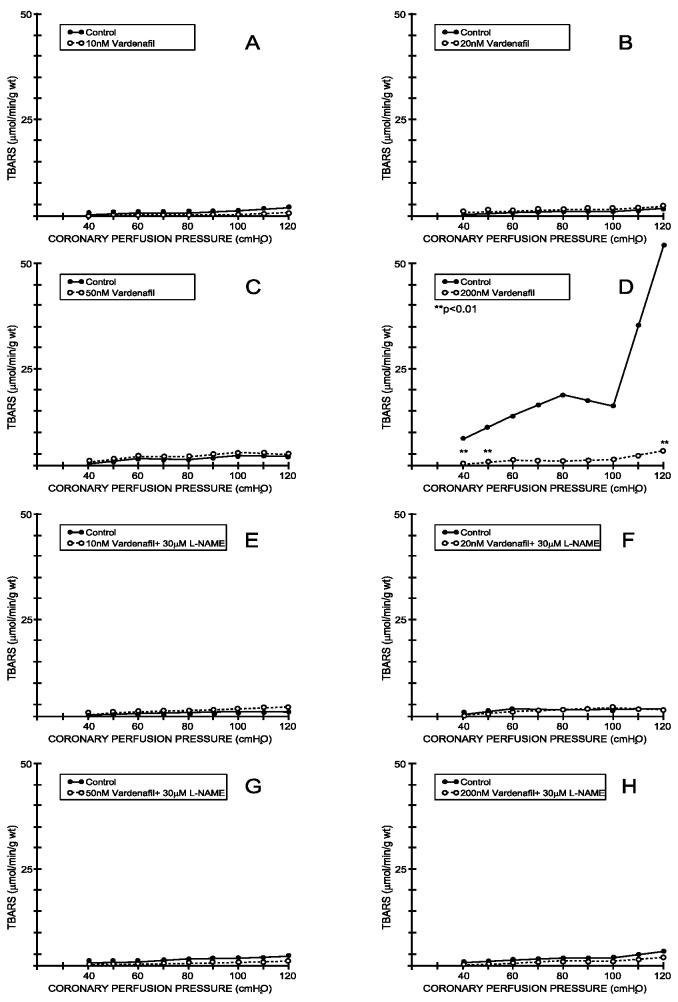
Effects of vardenafil and combination vardenafil + L-NAME on index of lipid peroxidation. (**A**) Vardenafil, 10 nM, (**B**) 20 nM, (**C**) 50 nM, (**D**) 200 nM, (**E**) Vardenafil + L-NAME, 10 nM + 30 μM, (**F**) Vardenafil + L-NAME, 20 nM + 30 μM, (**G**) Vardenafil + L-NAME, 50 nM + 30 μM, (**H**) Vardenafil + L-NAME, 200 nM + 30 μM.

**Table 1 medicina-59-00458-t001:** Echocardiography parameters measured groups treated with tadalafil and vardenafil.

	Tadalafil Group (n = 12)	Vardenafil Group (n = 12)
IVSd	0.19 ± 0.01	0.20 ± 0.01
LVIDd	0.71 ± 0.02	0.72 ± 0.03
PWd	0.23 ± 0.01	0.24 ± 0.01
IVSs	0.24 ± 0.01	0.23 ± 0.02
LVIDs	0.42 ± 0.02	0.43 ± 0.01
PWs	0.29 ± 0.01	0.30 ± 0.01
FS	39.15 ± 3.02	39.00 ± 2.34
EF	61.94 ± 3.21	62.28 ± 4.45

IVSd—end-diastolic interventricular septal thickness (cm); LVIDd—left ventricular internal diameter end diastole (cm); PWd—left ventricular end-diastolic posterior wall thickness (cm); IVSs—end-systolic interventricular septal thickness (cm); LVIDs—left ventricular internal diameter end systole; PWs—left ventricular posterior wall thickness (cm); FS—fractional shortening (%); EF—ejection fraction (%). Values are shown as mean ± SD.

## Data Availability

Not applicable.

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
