# Peer review of "Changes in Left Ventricular Ejection Fraction and Oxidative Stress after Phosphodiesterase Type-5 Inhibitor Treatment in an Experimental Model of Retrograde Rat Perfusion"

_medicina, 2023, doi:10.3390/medicina59030458_

Round 1
Reviewer 1 Report
In this study, Krivokapic et al. study the effects of vardenafil and tadalafil on global cardiac IR injury and conclude that both effectively augment LV function compared with placebo.
1. The figures are extremely hard to follow as letters (e.g. A, B) are doubled. This should be revised.
2. A significant, extensive English language editing is required as there are several grammatical flaws throughout the mansucript.
3. An important discrepancy is the fact that vardenafil, more so, tadalafil induce significant expression of nitro-oxidative stress markers (superoxide, TBARS). Generally, PDE5 inhibitors reduce MI size and protect against LV dysfunction. The authors need to carefully dissolve this significant discrepancy with the literature.
4. Some measurements show very high variability across study groups. E.g. in controls in tadalafil study are >10-times higher than in controls used in vardenafil study.
Author Response
R1
In this study, Krivokapic et al. study the effects of vardenafil and tadalafil on global cardiac IR injury and conclude that both effectively augment LV function compared with placebo.
- The figures are extremely hard to follow as letters (e.g. A, B) are doubled. This should be revised.
A: The all figures as well as Legends of figures are separated. Please see these parts of manuscript. Thank you.
- A significant, extensive English language editing is required as there are several grammatical flaws throughout the mansucript.
A: The manuscript is corrected by using English service provided my MDPI with native English lecturer. Please find the Certificate below.
- An important discrepancy is the fact that vardenafil, more so, tadalafil induce significant expression of nitro-oxidative stress markers (superoxide, TBARS). Generally, PDE5 inhibitors reduce MI size and protect against LV dysfunction. The authors need to carefully dissolve this significant discrepancy with the literature.
A: Corrected, Please see Manuscript. Thank you.
- Some measurements show very high variability across study groups. E.g. in controls in tadalafil study are >10-times higher than in controls used in vardenafil study.
A: Each measuring had its control measurement. So that is the reason for the differeces of controls. For us, the most important are changes in relation to its control, but not in relation to another control. Please see Figures and Legends
R2
This article evaluated the changes in the left ventricular ejection fraction and oxidative stress after phosphodiesterase type-5 inhibitors treatment in experimental model of retrograde rat perfusion. There are some issues in this manuscript that should be addressed as follows:
- Abstract:
- The subheadings shouldn’t be numbered.
A: Corrected. Please see Manuscript text.
- The subheading “Background” should be changed to “Background and objectives”.
A: Corrected. Please see Manuscript text.
- The total number of rats used in this study should be mentioned in the “Abstract”.
A: Corrected. Please see Abstract text.
- Introduction:
- The novel points in this study and how they differ from other studies should be explained in the introduction section.
A: Corrected, Please see Manuscript. Thank you.
- The aim of the study mentioned in the “Introduction” section should be similar to that mentioned in the “Abstract”.
A: Corrected. Please see Abstract text.
- Materials and methods:
- The total number of rats used in this study should be mentioned.
A: Corrected. Please see Section Methods.
- Page 2 Lines 88,89: A reference for the anesthesia method should be mentioned.
A: Corrected. Please see Section Methods.
- Page 3: A reference should be added to the method of retrograde perfusion of rat heart.
A: Corrected. Please see Section Methods.
- How did you know that the animals were acclimatized?
A: According to the definition of acclimatization: “Acclimation refers to the process/period during which newly arrived research animals are allowed to fully recover from shipping and adjust to new surroundings, feed, light/dark cycles, cage/pen mates, and personnel prior to being used on research, teaching, or testing protocols. It also provides a period for physiologic, psychological, and nutritional stabilization prior to use.” We are strictly guided to the Good Laboratory Practice guidelines and The Guide for the Care and Use of Laboratory Animals and the Guide for the Care and Use of Agricultural Animals in Agricultural Research and Teaching recommend a period of stabilization and acclimation for newly arrived animals. The next references explained that in detail:
- Obernier JA, Baldwin RL. Establishing an appropriate period of acclimatization following transportation of laboratory animals. ILAR J. 2006;47(4):364-9
- NRC [National Research Council] 1996.Guide for the Care and Use of Laboratory Animals.
- FASS [Federation of Animal Science Societies] 1999. Guide for the Care and Use of Agricultural Animals in Agricultural Research and Testing.
- NRC [National Research Council] 2006. Guidelines for the Humane Transportation of Research Animals
- Aguila HN, Pakes SP, Lai WC, Lu YS. The effect of transportation stress on splenic natural killer cell activity in C57Bl/6J mice. Lab Anim Sci. 1988 38: 148-151
- Landi MS, Kreider JW, Lang CM, Bullock LP. 1982. Effects of shipping on the immune function in mice. Am J Vet Res 43: 1654-1657
- McGlone, JJ et al. Shipping stress and social status effects on pig performance, plasma cortisol, natural killer cell activity, and leukocyte numbers. J Anim Sci 1993 Apr 71(4):888-96.
- Hicks TA et al. Behavioral, endocrine, immune, and performance measures for pigs exposed to acute stress. J Anim Sci 1998 Feb 76(2):474-83.
- Murata H, et al. Influence of truck transportation of calves on their cellular immune function. Nippon Juigaku Zasshi 47: 823-827.
- Tuli Js et al. Stress measurements in mice after transportation. Lab Anim 1995 29:132-138.
- Bean-Knudsen DE and Wagner, JE. Effect of shipping stress on clinicopathologic function in F344/N rats. Am J Vet Res 1987 Feb. 48(2): 306-8.
- Rowland RT, et al. Transportation or noise is associated with tolerance to myocardial ischemia and reperfusion injury. J Surg Res 89: 7-12.
- Knowles TG, et al. Effects on cattle of transportation by road for up to 31 hours. Vet Rec 145: 575-582.
- Capdevila, S et al Acclimatization of rats after ground transportation to a new animal facility. Lab Anim 2007, Apr 41(2):255-61.
- The housing conditions of the animals should be mentioned.
A: The answer is the same as on previous question. Please see above. Also, we added it into the Section Method with subheading Animals. Please see.
- The exact source and CAT numbers of the kits and chemicals that were used in this study should be mentioned.
A: Markers of oxidative stress were measured basic methods and spectrofotometricaly, without using any kits for Elisa. Please see Section Method.
- Other markers for oxidative stress can be measured to add value to the study such as the total antioxidant capacity.
A: Unfortunately, in this moment samples were not available. Based on this comment, we want to remind the reviewers that we measured the exact concentration of mostly dominant biomarkers. Total antioxidant status (TAS) gives information about all of the antioxidants in the organism, while index of lipid peroxidation (TBARS) is a size of lipid peroxidation and marker used to assess lipid peroxidation due to increased oxidative stress.
- Statistical analysis should be re-written to identify the exact statistical tools used for analysis of the results of the present study.
A: Corrected, Please see Section Statistical Methods.
- Results:
- Error bars should be added to figures 2-8.
A: Corrected, Please see Figures.
- A collective diagram summarizing the main findings of this study is recommended.
- Discussion: The discussion should be re-written to include more details about analysis of the results of the present study with addition of more recent references.
A: Corrected, Please see Manuscript. Thank you.
- Conclusion: The possible implication of the findings of the present study in the clinical settings should be mentioned.
A: Corrected, Please see Manuscript. Thank you.
- General comments:
- The manuscript should be revised by English-naïve speaker to improve the quality of the language.
English lecture was done in collaboration with native speakers from MDPI. Please see Certificate.
- The manuscript should be checked regarding the grammatical errors and plagiarism.
English lecture was done in collaboration with native speakers from MDPI. Please see Certificate.

Reviewer 2 Report
Reviewer comments:
This article evaluated the changes in the left ventricular ejection fraction and oxidative stress after phosphodiesterase type-5 inhibitors treatment in experimental model of retrograde rat perfusion. There are some issues in this manuscript that should be addressed as follows:
· Abstract:
1. The subheadings shouldn’t be numbered.
2. The subheading “Background” should be changed to “Background and objectives”.
3. The total number of rats used in this study should be mentioned in the “Abstract”.
· Introduction:
1. The novel points in this study and how they differ from other studies should be explained in the introduction section.
2. The aim of the study mentioned in the “Introduction” section should be similar to that mentioned in the “Abstract”.
· Materials and methods:
1. The total number of rats used in this study should be mentioned.
2. Page 2 Lines 88,89: A reference for the anesthesia method should be mentioned.
3. Page 3: A reference should be added to the method of retrograde perfusion of rat heart.
4. How did you know that the animals were acclimatized?
5. The housing conditions of the animals should be mentioned.
6. The exact source and CAT numbers of the kits and chemicals that were used in this study should be mentioned.
7. Other markers for oxidative stress can be measured to add value to the study such as the total antioxidant capacity.
8. Statistical analysis should be re-written to identify the exact statistical tools used for analysis of the results of the present study.
· Results:
1. Error bars should be added to figures 2-8.
2. A collective diagram summarizing the main findings of this study is recommended.
· Discussion: The discussion should be re-written to include more details about analysis of the results of the present study with addition of more recent references.
· Conclusion: The possible implication of the findings of the present study in the clinical settings should be mentioned.
· General comments:
1. The manuscript should be revised by English-naïve speaker to improve the quality of the language.
2. The manuscript should be checked regarding the grammatical errors and plagiarism.
Author Response

(The authors gave the same response as above.)

Round 2
Reviewer 1 Report
The authors reponded to all critiques.
Reviewer 2 Report
The authors had appropriately addressed most of my comments